# Inducible Nitric Oxide Synthase (*iNOS*) Mediates Vascular Endothelial Cell Apoptosis in Grass Carp Reovirus (GCRV)-Induced Hemorrhage

**DOI:** 10.3390/ijms20246335

**Published:** 2019-12-16

**Authors:** Bo Liang, Jianguo Su

**Affiliations:** 1Department of Aquatic Animal Medicine, College of Fisheries, Huazhong Agricultural University, Wuhan 430070, China; liangbo@webmail.hzau.edu.cn; 2Laboratory for Marine Biology and Biotechnology, Pilot National Laboratory for Marine Science and Technology, Qingdao 266237, China

**Keywords:** inducible nitric oxide synthase, S-methylisothiourea sulfate, grass carp reovirus, hemorrhage, apoptosis

## Abstract

Hemorrhage is one of the most obvious pathological phenomena in grass carp reovirus (GCRV) infection. The etiology of GCRV-induced hemorrhage is unclear. We found inducible nitric oxide synthase (*iNOS*) may relate to viral hemorrhage according to the previous studies, which is expressed at high levels after GCRV infection and is related to apoptosis. In this study, we aimed to investigate the mechanism of *iNOS* on apoptosis and hemorrhage at the cell level and individual level on subjects who were infected with GCRV and treated with S-methylisothiourea sulfate (SMT), an *iNOS* inhibitor. Cell structure, apoptosis rate, and hemorrhage were evaluated through fluorescence microscopy, Annexin V-FITC staining, and H&E staining, respectively. Cell samples and muscle tissues were collected for Western blotting, NO concentration measure, caspase activity assay, and qRT-PCR. *iNOS*-induced cell apoptosis and H&E staining showed that the vascular wall was broken after GCRV infection in vivo. When the function of *iNOS* was inhibited, NO content, apoptosis rate, caspase activity, and hemorrhage were reduced. Collectively, these results suggested *iNOS* plays a key role in apoptosis of vascular endothelial cells in GCRV-induced hemorrhage. This study is the first to elucidate the relationship between *iNOS*-induced cell apoptosis and GCRV-induced hemorrhage, which lays the foundation for further mechanistic research of virus-induced hemorrhage.

## 1. Introduction

Hemorrhage is a common pathological phenomenon. It could happen anywhere in the body where blood vessels are damaged. It can develop as a result of traumatic injury, fungal infection (such as *Alternaria* species, *Cladosporium* species, and *Curvularia* species) [1,2], bacterial infection (such as *Pseudomonas fluorescens*, and *Aeromonas hydrophila*) [3,4], viral infection (such as hantavirus, ebolavirus, and grass carp reovirus) [5,6,7], and some unknown causes. Hemorrhage is one of the most observable pathological phenomena in grass carp reovirus (GCRV) infection [8,9]. GCRV can cause hemorrhage disease in grass carp (*Ctenopharyngodon idella*) [10], black carp (*Mylopharyngodon piceus*) [11], rare minnow (*Gobiocypris rarus*) [12], etc. GCRV is a highly pathogenic viral agent that results in huge losses to the grass carp industry [13]. GCRV is icosahedron dsRNA virus without envelope, a member of the genus *Aquareovirus* and the family *Reoviridae*. GCRV is divided into three types (GCRV-I, GCRV-II and GCRV-III), and GCRV-II is the most prevalent strain in recent years in China [14]. In the present study, the GCRV097 strain (GCRV-II) is employed.

Previous research has shown that inducible nitric oxide synthase (*iNOS*) may be related to hemorrhage disease [15,16]. *iNOS* is one of four isoforms in the family of nitric oxide synthases (NOSs). Neuronal NOS (nNOS) exists in nervous tissue [17]. Endothelial NOS (*eNOS*) is in the endothelium [18]. Inducible NOS (*iNOS*) appears in the immune system and cardiovascular system [19]. Bacterial NOS (*bNOS*) can be found in various Gram-positive bacteria [20]. *nNOS* and *eNOS* exist in the neurocyte and endothelium, respectively, and their activities are switched by Ca^2+^ concentrations. In contrast, the expression of *iNOS* is determined by the de novo synthesis of *iNOS* mRNA and protein in various tissue and cell types [21,22]. *iNOS* significantly affects the responses of 874 genes to cytokines and bacteria. Previous studies showed *iNOS* regulated about 200 genes and led to at least a two-fold change in expression level [23]. *iNOS* is regulated by cytokines, viral products, oxygen tension, cell–cell contact, and various antibiotics, but it is not regulated by Ca^2+^ concentrations. Furthermore, it can produce more nitric oxide (NO) than *eNOS* and *nNOS*. NOSs catalyze the arginine to produce NO—an important signaling molecule [24]. NO regulates the normal physiological activity and can also be cytotoxic. The function of NO as regulatory or cytotoxic is determined by the magnitude as well as the duration [25]. NO diffuses into vascular smooth muscle cells and activates the cyclic guanosine monophosphate pathway to elicit vasorelaxation, which helps lower blood pressure. A lack of NO in the kidney has been proposed as a main cause of systemic hypertension [26]. NO has multiple functions in the immune system, such as antimicrobial activity, anti-tumor, tissue-damaging effect, anti-inflammatory-immunosuppressive effect, and more. NO also impedes the adhesion of platelets to endothelium, thus inhibiting healing of the injury of the vessel [23]. The massive quantity of NO produced by *iNOS* acts as a cytotoxic agent. NO transforms into peroxynitrite (ONOO^-^), which diffuses through the membranes and causes damage as it is a conjugate acid [27,28]. A smaller amount of ONOO^-^ activates the process of cell apoptosis via inducing damage of the mitochondrion to release cytochrome C [29,30]. Apoptosis is a type of programmed cell death and can be activated through the extrinsic pathway, the intrinsic pathway, and the perforin pathway [31]. Viral infection also can induce cell apoptosis. For example, Dengue virus infects the human microvascular endothelial cells and the viral protease interacts with NF-κB inhibitor. Additionally, p50 and p65 translocate into the nucleus and activate downstream genes. Subsequently, caspase-8 and caspase-9 are activated and the cell apoptosis is developed [32]. Caspase-3 and caspase-9 are aspartate-specific cysteinyl protease and major proteins in the process of apoptosis [33]. Some aquatic viruses, such as Cyprinid herpesvirus 3 and Spring viremia of carp virus, can active caspases to induce apoptosis, and can also up-regulate the expression of *iNOS* [34].

The coagulation system and anticoagulation system are important components in the blood circulation system [35,36]. The coagulation system is immediately activated after the vascular endothelium is damaged [37]. Platelets bind directly to the underlying collagen, tissue factor pathway and contact activation pathway are also activated, and prothrombin and fibrinogen are subsequently activated. Stable fibrin clots then bind to the injury site to block bleeding [38]. The anticoagulation system balances the blood circulation system by counteracting the coagulation system [39]. Disorders of coagulation and anticoagulation can result in hemorrhage, thrombosis, or bruising [40,41]. Virus infection also affects coagulation and anticoagulation in the circulatory system. For example, disordered anticoagulation of patients caused by Epstein–Barr virus infection leads to intravascular coagulation [42]. Based on previous studies, we chose several factors in the present hemorrhage disease caused by a viral infection, including the coagulation factors: *kininogen*-1 (*kng-1*), *f2*, *f3a*, *f3b* and *f10*, as well as the anticoagulation factors: *serine protease inhibitor* (*serpin*) *b1*, *c1*, *d1*, *f1*, *f2b*, and *g1* [43,44].

Accordingly, in this study, we aimed to investigate the relationship between *iNOS*, cell apoptosis, and GCRV-induced hemorrhage in order to explain the mechanism of hemorrhage after GCRV infection.

## 2. Results

### 2.1. iNOS Relates to Hemorrhage Symptom Caused by GCRV Infection

To search the gene that relates with GCRV-induced hemorrhage, the previous research data were employed to carry out a cross-comparison analysis and screen the co-changing genes in multiple organs [15,45]. These genes were identified with BLAST, and an advanced analysis technique was used to identify the gene that relates to hemorrhage (Figure 1). The result indicated that *iNOS* may play a role in GCRV-induced hemorrhage.

### 2.2. iNOS Induces Cell Apoptosis

Based on the previous finding, we questioned whether grass carp *iNOS* could induce cell apoptosis similarly to mammalian *iNOS* or not. The grass carp *iNOS* was cloned and FHM cells were transfected with overexpression vector, we found *iNOS* could over-express in FHM cells (Figure 2A). The plasmid we used could express the green fluorescent protein independently, allowing us to view the status of cells by fluorescence microscopy. The control group showed the complete structures of the cell nucleus and the cell membrane. The samples were collected at 12, 24, 48, and 72 h post transfection. There was no change at 12 h. But from 24 h to 72 h, the nuclei condensed and fragmented. The cells were broken and formed apoptosis bodies (Figure 2B). The activities of caspase-3 and caspase-9 were assayed by Caspase Activity Assay Kits. Caspase-3 and caspase-9 were activated by the overexpression of *iNOS* via the transfection of pCICE in FHM cells. Caspase-3 was activated at 24 h post transfection, but caspase-9 did not activate. At 48 h and 72 h, the activities of caspase-3 and caspase-9 were significantly up-regulated (Figure 2C).

### 2.3. SMT Inhibits Cell Apoptosis Caused by GCRV Infection

To investigate whether SMT, a selective inhibitor of *iNOS*, inhibits cell apoptosis caused by GCRV infection via inhibiting *iNOS*. FHM cells were treated with different concentrations of SMT after GCRV infection. After a 48 h treatment with GCRV and SMT, FHM cells demonstrated evident changes in different SMT concentration treatments. From 0 to 1 mM, the cell lesions were attenuated as the SMT concentration increased and the dead cells and cell debris decreased (Figure 3A). Apoptotic cells were detected by flow cytometry following annexin V-FITC staining in different SMT concentrations. The fluorescent signal of FITC was slacked as the SMT concentration augmented. The percent of apoptotic cells was 95.86% in 0 μM SMT, 31.49% in 1 μM, 25.36% in 10 μM, 29.06% in 0.1 mM and 19.78% in 1 mM (Figure 3B).

### 2.4. SMT Suppresses the Function of iNOS in Vivo

Given that SMT can inhibit cell apoptosis in vitro after GCRV infection, we conjectured that SMT can also inhibit cell apoptosis in vivo after GCRV infection. Rare minnow were treated with SMT after GCRV infection, we found SMT did not affect *iNOS* at the RNA level and the protein level. The RNA level and the protein level of *iNOS* did not change in the GCRV group compared to the GCRV+SMT group (Figure 4A,B). The concentration of NO in the SMT+GCRV group was significantly lower than that in the GCRV group (Figure 4C). This indicated that SMT significantly inhibits the activity of *iNOS* in vivo. The activities of caspase-3 and caspase-9 were significantly up-regulated in the SMT+GCRV group and the GCRV group than those in the Blank group, and the activities of caspase-3 and caspase-9 in the GCRV+SMT group were significantly lower than those in the GCRV group. The activities of caspase-3 and caspase-9 were significantly down-regulated upon reducing the concentration of NO (Figure 4D).

### 2.5. The Hemorrhage is Reduced When Inhibiting the Function of iNOS

To verify the relationship between hemorrhage—an obvious pathological phenomenon after GCRV infection—and the overexpression of *iNOS* after GCRV infection, rare minnow were treated with SMT after GCRV infection. Hemorrhagic symptoms appeared on day 5 post-GCRV injection in the GCRV group and GCRV+SMT group. In the GCRV group, the gloss on the body surface was fading and the muscle had acute hemorrhage. In the GCRV+SMT group, the gloss on the body surface was fading, but by a lesser degree than that of the GCRV group, and the muscle hemorrhage was also less than that in the GCRV group. In the blank group, the gloss on the body surface was normal and there was no subcutaneous hemorrhage (Figure 5A). Based on H&E stain analysis of muscle sections, the gap among the muscle cells became larger in the GCRV group and GCRV+SMT group compared to the blank group, likely edema, and the blood cells infiltrated in the gap (Figure 5B). The percent of the blood cell infiltration was calculated by ImageJ (1.51k, National Institutes of Health, Bethesda, MD, USA): GCRV group was 1.14930%, GCRV+SMT group was 0.40659%, and blank group was 0.00929%. The black dots showed the blood cells in muscle tissue (Figure 5C). Rare minnow died starting from the fifth day after GCRV infection and all died by the seventh day (Figure 5D).

### 2.6. iNOS Breaks the Vascular Wall Via Activating Cell Apoptosis

To determine the status of vessels in different treatments, H&E stained sections were made. We found the capillary in the blank group is intact with a clear capillary wall and no damage. The red blood cells passed in a line through the capillary. In the GCRV+SMT group, most of the vascular wall was still complete with partial blood cell infiltration. In the GCRV group, the vascular wall was damaged and not clear. The blood cells that were located in the tissue gap were from the damaged vessel (Figure 6).

### 2.7. iNOS Activates Both the Coagulation and the Anticoagulation Systems

The changing expressions of coagulation factors and anticoagulation factors are important in hemorrhage. To investigate the effect of *iNOS* on these factors, the coagulation factors and anticoagulation factors were detected by qRT-PCR. The anticoagulation factors *serpinc1*, *serpinf2*, and *serping1* were significantly up-regulated in the GCRV group (Figure 7A), while the anticoagulation factors *serpinb1*, *serpind1*, and *serpinf1* had no significant change (Appendix A). The coagulation factors *f2*, *f3a*, and *kng*-*1*, were significantly up-regulated in the GCRV group (Figure 7B), while the coagulation factors *f10* and *f3b* showed no significant change (Appendix A).

## 3. Discussion

Viral hemorrhage is one of the severe pathological phenomena in GCRV infection [46]. The etiology of the GCRV-induced hemorrhage is still not clear. This study found that *iNOS* could lead to GCRV-induced hemorrhage. High quantities of NO generated by *iNOS* defend against viral infection [47]. A large quantity of NO activates caspase-3 and caspase-9 and induces cell apoptosis by releasing cytochrome C via mitochondrial apoptosis [48]. The aim of this study was to investigate the effect of *iNOS* on cell apoptosis and hemorrhage during GCRV infection.

Following the overexpression of *iNOS* in FHM cells, cell apoptosis occurred and the activities of caspase-3 and caspase-9 were significantly increased. These observations aligned with previous literature, which reported that the *iNOS* could generate high quantities of NO and induce cell apoptosis [30,49]. Therefore, grass carp *iNOS* was confirmed to have the same function as mammalian *iNOS*.

To investigate the inhibitor of *iNOS*, FHM cells were infected with GCRV, and a significant cytopathic effect and high rate of cell apoptosis were observed. When treated with SMT—a competitive inhibitor of *iNOS*, which has no direct effect on GCRV replication—coagulation factors and anticoagulation factors (Appendix A) [50], the rate of cell apoptosis decreased and the cytopathic effect was attenuated. These observations demonstrated that SMT can reduce cell apoptosis by inhibiting *iNOS*.

Rare minnow were infected with GCRV, we observed obvious symptoms: vascular wall was broken and blood cells infiltrated into muscle tissue [51], the gloss on the body surface faded, the concentration of NO and activities of caspase-3 and caspase-9 significantly increased. Additionally, these effects were significantly attenuated by inhibition of *iNOS* via SMT treatment. These results indicated that *iNOS* reduces GCRV-induced hemorrhage. *iNOS* produced high quantities of NO to activate caspases and induce vascular endothelial cells apoptosis, and release blood cells out of the vessel. The vascular endothelial cells apoptosis resulted in hemorrhage development. This result is similar to the previous literature—dengue viral protease that interacts with NF-κB inhibitor α/β results in endothelial cell apoptosis and hemorrhage development [32].

Comprehensive comparative analyses of the expressions of key coagulation factors and key anticoagulation factors show that both the coagulation system and anticoagulation system are activated in GCRV infection, and the up-regulated expressions of anticoagulation factor genes were more prominent than those of coagulation factor genes. Therefore, anticoagulation may be more dominant in rare minnow during GCRV infection, which is the opposite in grass carp infected with GCRV [43]. NO also affects the coagulation disorder, it impedes the adhesion of platelets to the endothelium to block the repair of the vessel injury [23]. The high level of NO in the GCRV group also enhanced hemorrhage. The survival curve showed no significant difference between the GCRV group and GCRV+SMT group, which indicated that SMT could not be used as a protective agent to improve the survival rate of GCRV infection, and hemorrhage is not the major cause of death in GCRV infection.

Based on previously published results and this study, we proposed a model to describe the molecular pathogenic process of GCRV-induced hemorrhage (Figure 8). GCRV infects the vascular endothelial cell and activates the expression of *iNOS* [15]. High quantities of NO are produced by *iNOS*, causing mitochondrial apoptosis and releasing cytochrome C [33]. Cytochrome C activates the caspase process and leads to vascular endothelial cell apoptosis [48]. Hemorrhage occurs following vascular endothelial cells apoptosis [32]. SMT inhibits the activity of *iNOS* to reduce NO [52].

## 4. Materials and Methods

### 4.1. Ethics Statement

The protocol of animal experiments was approved by the Animal Management and Ethics Committee, Huazhong Agricultural University. The approval number is HZAUFI-2019-005 (Approval date: 23 April 2019). This experiment was conscientiously abided by the ethical principles of animal welfare. All surgery was performed under MS-222 anesthesia to minimize suffering.

### 4.2. Animal Experiments

A total of 85 rare minnow—weight 0.928 ± 0.107 g, length 4.12 ± 0.218 cm—were bought from the Institute of Hydrobiology, Chinese Academy of Sciences. They were cultured in our laboratory with aerated freshwater at 28 °C and fed with red worm twice a day. A total of 25 rare minnow were used in the pre-test to determine the concentrations of S-methylisothiourea sulfate (SMT, a selective *iNOS* inhibitor, No. S0008, Beyotime Biotechnology, Shanghai, China) and GCRV in vivo by intraperitoneal injection [52]. A total of 60 rare minnow were randomly divided into 3 groups: Blank group, GCRV group (GCRV infection, GCRV 50 μL), and GCRV+SMT group (GCRV infection and SMT treatment, GCRV 50 μL, SMT 100 mg/kg). The GCRV titer is 2.56 × 10 RNA copies/µL. All samples were collected under MS-222 anesthesia when hemorrhage developed.

### 4.3. Cloning of iNOS, Plasmids Construction, Cell Culture, and Transfection

The full-length cDNA sequence of grass carp *iNOS* has been reported (GenBank accession number HQ589354). The full-length coding sequence of *iNOS* was amplified using PrimeSTAR^®^ Max DNA polymerase (Takara, Kusatsu, Japan) with specific primers (Table 1), which is 3243 bp. The plasmid pCMV-*iNOS*-3×Flag-CMV-eGFP (pCICE, Appendix A) was constructed using a One-Step Cloning Kit (ClonExpress^®^ MultiS One Step Cloning Kit, Vazyme Biotech Co., Ltd., Nanjing, China) by introducing the *iNOS* coding sequence into the pCMV-3×Flag-CMV-eGFP (pCCE).

Fathead minnow (FHM) cells, epithelial cells from fathead minnow and with an ATCC number of CCL-42 [53], were grown in M199 (Gibco, Carlsbad, CA, USA) supplemented with 10% fetal bovine serum (Gibco, Carlsbad, CA, USA). The cells were incubated at 28 °C in a 5% CO_2_ humidified atmosphere. FHM cells were transfected with 800 ng pCICE in a six-well plate (Thermo Scientific, Waltham, MA, USA). The control group was transfected with 800 ng pCCE. All of the vectors were transfected into FHM cells by using the FuGENE^®^6 Transfection Reagent (Promega, Madison, WI, USA) according to the manufacturer’s instructions of reagent.

### 4.4. GCRV Infection and SMT Inhibitor

FHM cells were plated in a six-well plate. The cells were infected with GCRV and treated with five different concentrations of SMT (0, 1 μM, 10 μM, 0.1 mM and 1 mM). These cells were imaged by the light microscope 48 h post-treatment and collected for analyzing the apoptosis rate by using flow cytometry after staining with Annexin V-FITC according to the product information of Annexin V-FITC Apoptosis Detection Kit (eBioscience, BMS500FI/20, Thermo Scientific, Waltham, MA, USA).

### 4.5. Fluorescence Microscopy

FHM cells were plated in glass-bottomed dishes (Nest, Wuxi, Jiangsu, China) and transfected with 800 ng pCICE in a six-well plate. The control group was transfected with 800 ng pCCE. The samples were collected at 12, 24, 48, and 72 h post-transfection. All of the cells on microscopic glass-bottomed dishes were fixed using 4% formaldehyde for 15 min at 37 °C, followed by incubation in 1 μg/mL Hoechst 33342 (Gibco, Carlsbad, CA, USA) for 15 min in the dark to stain the nuclei. Then, the stained cells were rinsed with phosphate-buffered saline (PBS). Images were taken with the Live Cell Imaging System (PerkinElmer, Waltham, MA, USA).

### 4.6. Protein Extraction and Western Blot

Rare minnow muscle tissue was homogenized in PBS with 1 mM phenylmethylsulfonyl fluoride (PMSF, Biosharp, Anhui, China). The homogenates were centrifuged at 13,800 g for 15 min at 4 °C. The supernatants were collected for Western blot. As for cell samples, the cells from the six-well plate at 48 h post-transfection were centrifuged at 1000 *g* for 5 min. The cells were resuspended in 100 μL PBS with 1 mM PMSF. The samples were frozen (−70 °C for 15 min) and thawed (room temperature) 3 times, and centrifuged at 3000 *g* for 30 min at 4 °C. The supernatants were collected for Western blot.

The supernatant liquid was mixed with 20 μL 5× SDS loading buffer and denatured at 100 °C for 10 min. The mixture was separated by 10% SDS-PAGE and transferred to nitrocellulose membranes (MilliporeSigma, Burlington, MA, USA). Membranes were put in fresh 5% skim milk dissolved in Tris-Buffered Saline Tween-20 (TBST) buffer for 2 h at 4 °C. They were then incubated with primary antibodies for 2 h at room temperature: anti-Flag (monoclonal, 1:5000, Abclonal Technology, Wuhan, China), anti-NOS2 (polyclonal, 1:2000, Abclonal Technology, Wuhan, China), and anti-β-tubulin (polyclonal, 1:5000, Abcam, Cambridge, United Kingdom), respectively. The membranes were then washed 3 times with TBST buffer on the horizontal shaker (15 min each time) and incubated with Ab-HRP for 45 min at room temperature. The signal was obtained by using the ECL reagents (K-12045, PEPTBIO, Wuhan, China) with Amersham Imager 600 (GE, MA, USA).

### 4.7. Caspase Activity Assay, NO Measure, and Histopathology

The cells and muscle tissue were collected for measuring the production of NO and assaying the activities of caspase-3 and caspase-9 by using NO Content Assay Kit (No. S0021, Beyotime Biotechnology, Shanghai, China) and Caspase Activity Assay Kits (No. C1115, and No. C1157, Beyotime Biotechnology, Shanghai, China) according to the manufacturer’s instructions, respectively. The samples of muscle tissue from all groups were placed in 4% paraformaldehyde. All of the tissue sections were made and imaged by the company Pinuofei Biotechnology (Wuhan, China).

### 4.8. qRT-PCR Assay

The muscle tissue of rare minnow from all treatment groups was homogenized in TRIZOL (Simgen, Hangzhou, China). Total RNAs were extracted with TRIZOL reagent, and converted to cDNA using the Reverse Transcription Kit HiScript II Q RT SuperMix for qPCR (+gDNA wiper) (Vazyme Biotech Co.,Ltd, Nanjing, China). All the cDNA concentrations were adjusted to 50 ng/μL. The qRT-PCR mixture consisted of 4 μL of cDNA sample, 3.1 μL of nuclease-free water, 7.5 μL of 2× SYBR Green master mix (No. BSB25L1B, BioEasy, Hangzhou, China), and 0.2 μL of each gene-specific primer (10 mM). The PCR cycling conditions were as follows: 1 cycle of 95 °C for 30 s, 45 cycles of 95 °C for 5 s, 60 °C for 30 s, 1 cycle of 95 °C for 15 s, 60 °C for 30 s, followed by dissociation curve analysis to verify the amplification of a single product. mRNA expression levels were normalized to the expression level of β-actin, and the data were analyzed using the 2^-△△*C*T^ method. The qRT-PCR primers are shown in Table 2.

### 4.9. Statistical Analysis

Statistical analyses and presentation graphics were carried out using the GraphPad Prism 6.0 software (GraphPad Software, San Diego, CA, USA). Results were presented as mean ± SD for at least three independent experiments. All data were subjected to one-way ANOVA, followed by an unpaired, two-tailed *t* test. A *p* value < 0.05 was considered to be a statistically significant difference (* *p* < 0.05, ** *p* < 0.01, *** *p* < 0.001).

## 5. Conclusions

In conclusion, the hemorrhage and vascular wall injury were observed after GCRV infection, and these pathological phenomena were both attenuated when the function of *iNOS* was inhibited. Therefore, *iNOS*-mediated vascular endothelial cell apoptosis plays a critical role in GCRV-induced hemorrhage. This study is the first to explain the relationship between *iNOS*-induced cell apoptosis and GCRV-induced hemorrhage; it provides a novel insight and lays a foundation for further mechanistic research of virus-induced hemorrhage.

## Figures and Tables

**Figure 1 ijms-20-06335-f001:**
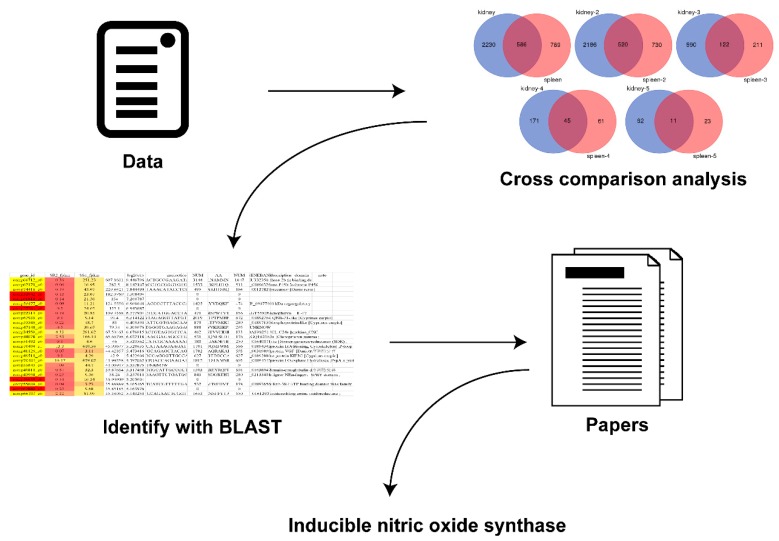
The process of bioinformatic analysis to find the hemorrhage-related gene, inducible nitric oxide synthase. The previous research data were used to perform a cross-comparison analysis and screen the co-changing genes in multiple organs. Then, the gene was identified with BLAST. Finally, inducible nitric oxide synthase (*iNOS*) was found.

**Figure 2 ijms-20-06335-f002:**
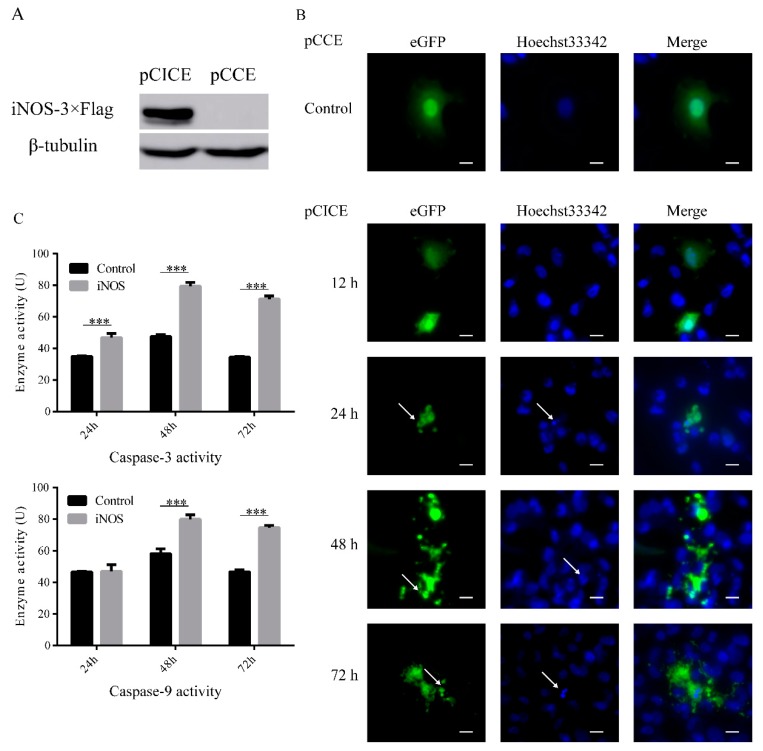
*iNOS*-induced cell apoptosis in FHM. (**A**) The protein overexpression of *iNOS* was detected by Western blot. FHM cells were transfected with 800 ng *iNOS*-3×Flag overexpression vector (pCMV-*iNOS*-3×Flag-CMV-eGFP, pCICE) in six-well plates. The control group was transfected with 800 ng empty vector (pCMV-CMV-eGFP, pCCE). Cell samples were collected after 24 h for Western blot. (**B**) *iNOS* overexpression induced cell apoptosis in FHM. FHM cells were transfected as (**A**) in glass-bottomed dishes. The samples were collected at 12 h, 24 h, 48 h, and 72 h and Incubated in 4% Paraformaldehyde and Hoechst 33342. The photos were imaged by fluorescence microscopy. There were no changes observed with the control group at 12 h. From 24 h to 72 h, the nuclei condensed and fragmented. The cells were broken and exploded into apoptosis bodies. As the white arrows show (scale 10 μm). (**C**) *iNOS* overexpression activated caspase-3 and caspase-9. FHM cells were transfected as (**A**) in the six-well plate. The samples were collected at 24 h, 48 h, and 72 h and the activity of caspase-3 and 9 were assayed by Caspase Activity Assay Kits. Data were analyzed as the mean ± SEM of the results from three independent experiments. *** *p* < 0.001.

**Figure 3 ijms-20-06335-f003:**
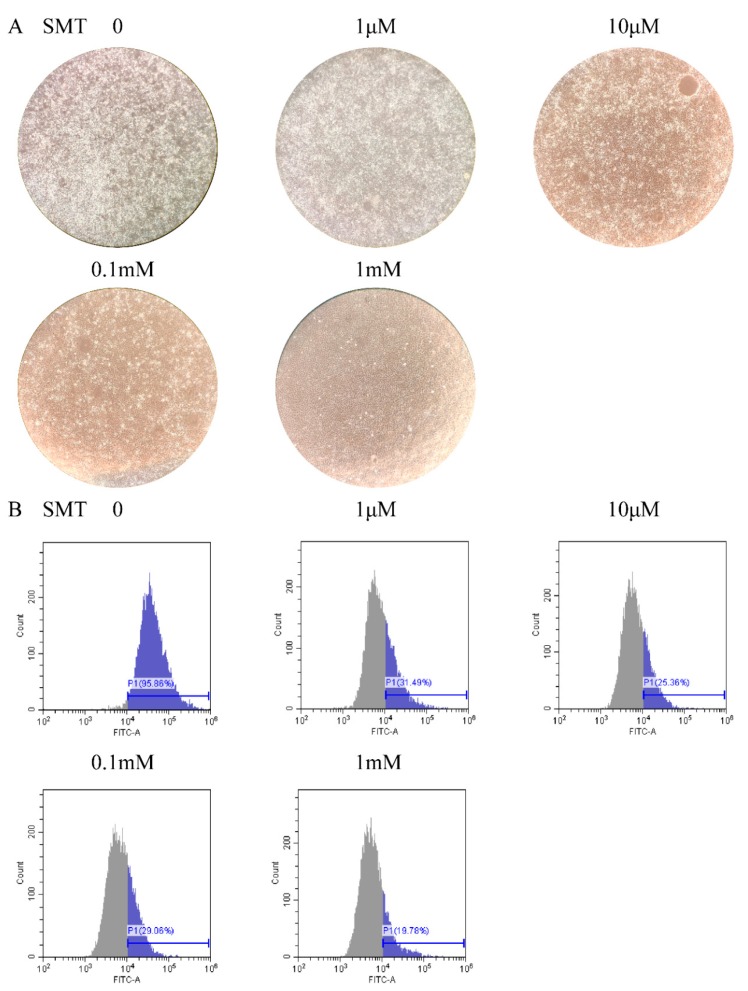
SMT inhibited cell apoptosis in GCRV infection. (**A**) The cytopathic effect decreased as the SMT concentration increased. FHM cells were infected with GCRV and treated with different SMT concentrations in the six-well plate, imaged by the light microscope (Zoom 100×). The white dots are dead cells and cell debris. (**B**) The apoptosis rate decreased with the SMT concentration increased. FHM cells in (**A**) were collected and apoptosis cells were detected by flow cytometry following annexin V-FITC staining. The blue part shows the percent of apoptosis in samples.

**Figure 4 ijms-20-06335-f004:**
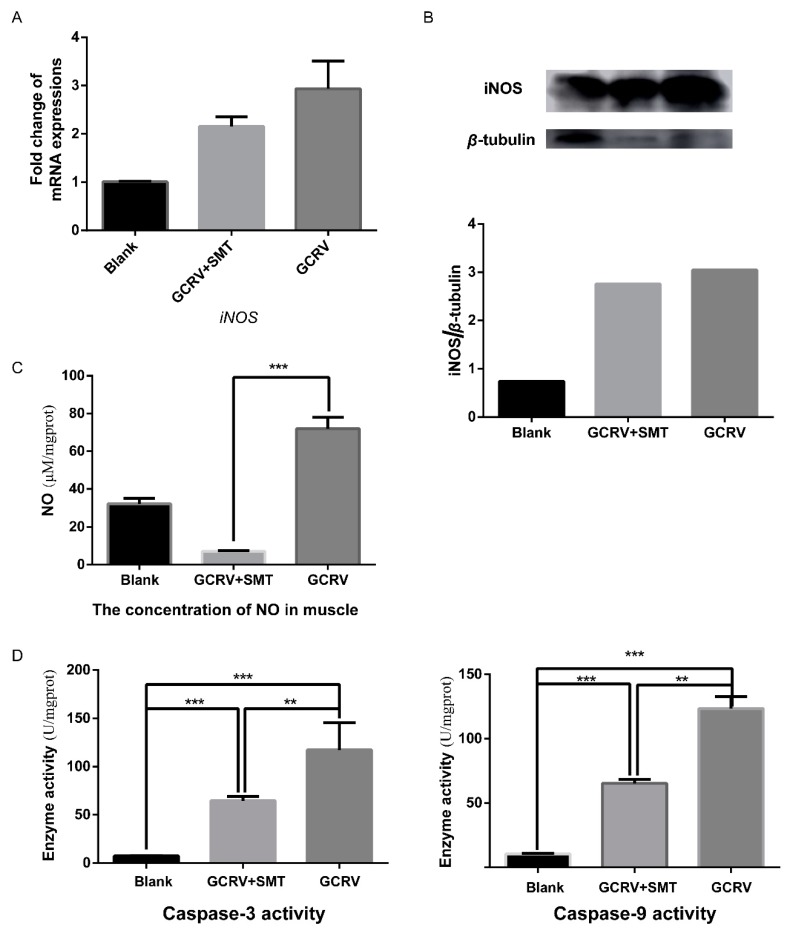
SMT did not reduce *iNOS* expressions but inhibited NO production and caspase activities in vivo. (**A**) SMT had no effect on the mRNA transcription of *iNOS* by qRT-PCR (*n* = 3). (**B**) SMT had no effect on the protein expression of *iNOS*. The expression of *iNOS* was detected by Western blot, and quantitated by ImageJ. (**C**) The NO content in muscle was significantly decreased in GCRV+SMT group VS. GCRV group. Quantification of NO production was assayed by NO Content Assay Kit (*n* = 3). (**D**) The activities of caspases were significantly decreased in GCRV+SMT group VS. GCRV group. The activity of caspase-3 and 9 were assayed by Caspase Activity Assay Kits (*n* = 3). (** *p* < 0.01, *** *p* < 0.001).

**Figure 5 ijms-20-06335-f005:**
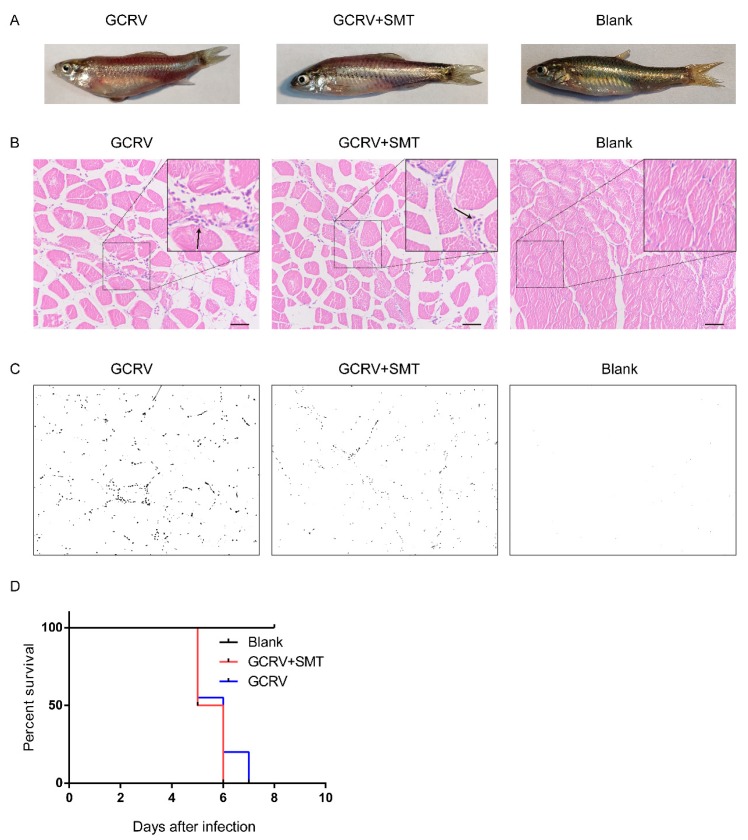
Hemorrhage status in individuals. (**A**) Hemorrhage was decreased when inhibiting the function of *iNOS* by SMT. Viewed from the surface of the rare minnow in different groups. The fading of the glossiness of the body surface and severe subcutaneous hemorrhage could be observed in the GCRV group. The slight fading of the glossiness of the body surface and minor subcutaneous hemorrhage in the GCRV+SMT group. The blank group has a normal body color and no subcutaneous hemorrhage. (**B**) H&E-stained sections show the infiltration of blood cells in muscle in different treatments. The black arrows show the blood cells (scale 50 μm). (**C**) The H&E-stained sections from (**B**) were analyzed with ImageJ image-analysis software to calculate the percent of blood cell infiltration. Black dots show the blood cells in muscle tissue. (**D**) Rare minnow died from the fifth day after GCRV infection and all died on the seventh day.

**Figure 6 ijms-20-06335-f006:**
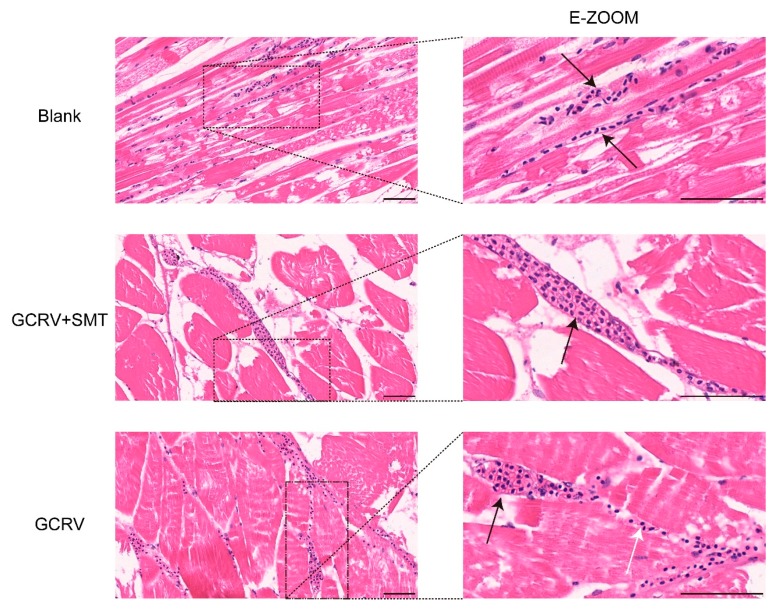
The vascular injury was inhibited when inhibiting the function of *iNOS* by SMT. H&E-stained sections of the vessel in different treatments (scale 50 μm). The vascular wall was broken and the infiltration of blood cells could be observed in the GCRV group. The vascular wall was complete in GCRV+SMT and blank. The black arrow shows the vessel and the white shows hemorrhage.

**Figure 7 ijms-20-06335-f007:**
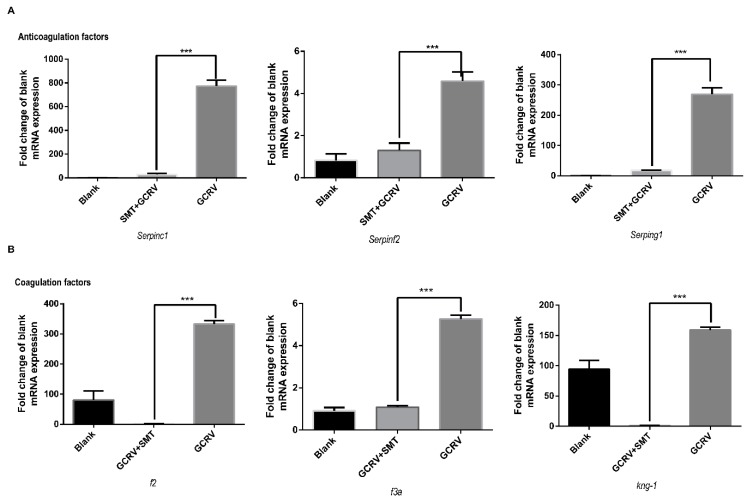
Some coagulation factors and anticoagulation factors were inhibited when suppressing the function of *iNOS* in muscle. (**A**) Anticoagulation factors serpinc1, serpinf2, and serping1 are significantly up-regulated in the GCRV group VS. GCRV+SMT (*n* = 3). (**B**) Coagulation factors f2, f3a, and kng-1 are significantly up-regulated in the GCRV group VS. GCRV+SMT (*n* = 3). (*** *p* < 0.001).

**Figure 8 ijms-20-06335-f008:**
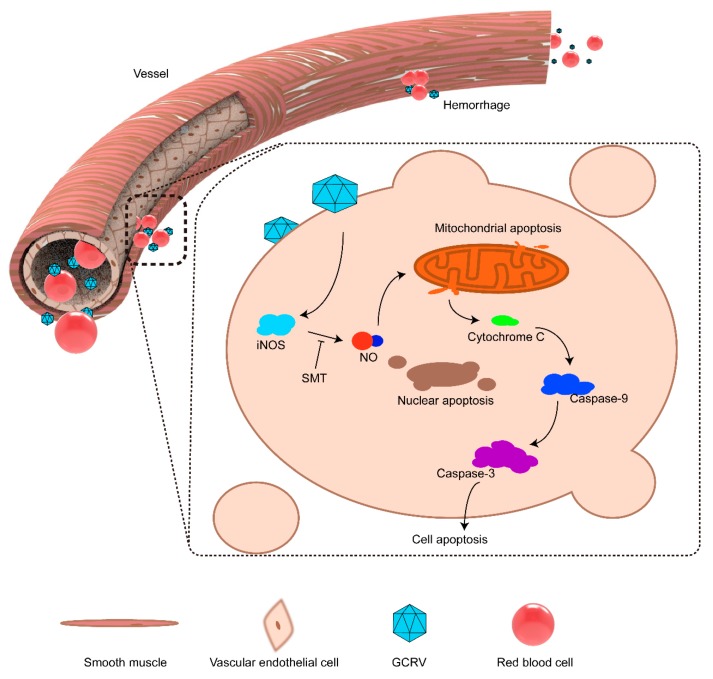
The proposed model for the mechanism of *iNOS*-induced endothelial cell apoptosis and hemorrhage. GCRV infects the vascular endothelial cell and activates the expression of *iNOS*. A large quantity of NO was produced by *iNOS* and leads mitochondrial apoptosis to release cytochrome C. The cytochrome C activates the caspase process and leads the vascular endothelial cell apoptosis. The hemorrhage happened when vascular endothelial cells apoptosis. SMT can inhibit the activity of *iNOS* to produce NO.

**Table 1 ijms-20-06335-t001:** The primers for amplifying *iNOS*.

Primer Name	Sequence (5′ → 3′)
NOS1F	CGTCAGATCGGAATTCGGTACCAATGGGGAACCAGGCCACTAAAGACA
NOS1R	GCTGCAAACTTATCGTCCACTGCGTGAG
NOS2F	CGCAGTGGACGATAAGTTTGCAGCACT
NOS2R	GTCAGCCCGGGATCCTCTAGAGGATCCGAAGATGTCCTCATGGTACCGC

**Table 2 ijms-20-06335-t002:** The primers for qRT-PCR.

Gene Name	Primer Name	Sequence (5′ → 3′)	Accession Number
*Serpinb1*	Sb1F	TCTCTGATGTTCACTCTCGGTTTG	MN326820
	Sb1R	TCTTTGATTTTATTCTCCGTCTGC	
*Serpinc1*	Sc1F	TCGATAAAGGAGAAGACATCTGACC	MN326821
	Sc1R	GCTGGAAGGACTCATTAAAAAGTGTAG	
*Serpind1*	Sd1F	GCAGAACTATGACTTGATTGACCAC	MN326822
	Sd1R	TGATAAAGGCATGAATCCGATGT	
*Serpinf1*	Sf1F	CTCTCCAATACTCTCCACGCAGT	MN326823
	Sf1R	CGCTTCCAACCATTCAGACATAC	
*Serpinf2b*	Sf2bF	ACCTTGATACACTCTGTGATGGGG	MN326824
	Sf2bR	GGAAATTATGACATTGGGCTGTT	
*Serping1*	Sg1F	TTCCTCTCAGACACTGTTGGTGG	MN326825
	Sg1R	GAAGATCAAGTTGGATTTGGGGT	
*iNOS*	NOSF	CAGGTCTGAAAGGGAATCCTATGA	MN326826
	NOSR	TTGGGTTGTCCAGTCTGCCTAG	
*Kng-1*	Kng1F	ACCGCTCATTCATTCTTTGGC	MN380324
	Kng1R	CCTGGTGCAGTTGCTCTTCTC	
*F2*	F2F	GGACCTCAAACCCCACAAATC	MN380325
	F2R	CCCACGACACAATGCCAATC	
*F3a*	F3aF	CAGGCCTACCTGCCCTTCC	MN380326
	F3aR	TTTTTCGCTCGCTTTCTTCG	
*F3b*	F3bF	TTTTCGGTGCTTTTTCTCGC	MN380327
	F3bR	TCTGTTTGTCCCTACCAACTCTTG	
*F10*	F10F	AGTAAAGTTGCCACACAGGGAGC	MN380328
	F10R	CAGTTGGGTAGTTGTAAGAATGTAAAGG	
*β-actin*	ACTF	AATTGTCCGTGACATCAAAGAGAA	MN562268
	ACTR	GATACCGCAAGATTCCATACCC	
*VP4*	VP4F	CGAAAACCTACCAGTGGATAATG	ADC81088
	VP4R	CCAGCTAATACGCCAACGAC

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
