# Peer review of "Inducible Nitric Oxide Synthase (iNOS) Mediates Vascular Endothelial Cell Apoptosis in Grass Carp Reovirus (GCRV)-Induced Hemorrhage"

_ijms, 2019, doi:10.3390/ijms20246335_

Round 1

Reviewer 1 Report

Please see comments and suggestions in attached PDF file.

Reviewer 2 Report

General comments

Grass carp reovirus (GCRV) is a pathogen that causes hemorrhagic disease in grass carp. It is considered one of the the most important viral diseases of carp. This paper is aimed to get further insight into the role of inducible nitric oxide synthase (iNOS) in the appearance of hemorrhagic symptoms induced by GCRV. The work is well presented and contains novel and interesting results. On the minus side, I feel that the objectives of this work are not clearly defined, being at some point more a descriptive study. The Discussion section is also lacking some comparison to other viruses of fish/carp causing hemorrhages as well as some discussion on what has been previously reported that would link the production of iNOS to virus-induced pathogenesis. My main concern is that the effect of the iNOS inhibitor (SMT) on GCRV replication is not shown in this study.

Specific comments

Results: I believe that every section should have a couple of introductory sentences to provide the background and the rationale of the experiments.

2.3. Does the iNOS inhibitor SMT have any direct effect on GCRV replication? This point must be checked in order to draw conclusions from figures 3-6.

In Figures 5/6 an analysis of the effect of iNOS overexpression on vascular damage in muscle would strengthen authors´conclusions.

Figure 6: I´m not an expert on hemorrhagic injuries but the images “GCRV” and “GCRV+SMT” do not seem significantly different.

Figure 7: Nothing is said about the effect of SMT treatment on the coagulation and anticoagulation factors. This is also absent in the Discussion. Was SMT expected to have a differential effect on the coagulation and the anticoagulation factors?

Discussion

Information on other viruses causing hemorraghic symptoms in fish (SVCV, VHSV) is missing in the discussion, along the possible induction of apoptosis / iNOS in other viral diseases of fish. The following reference is suggested for citation: Vet Microbiol. 2015 Mar 23;176(1-2):19-31. doi: 10.1016/j.vetmic.2014.12.012. Differential effects of alloherpesvirus CyHV-3 and rhabdovirus SVCV on apoptosis in fish cells. Miest J, Adamek M, Pionnier N, Harris S, Matras M, Rakus K, Irnazarow I, Steinhagen D, Hoole D.

Line 144: “hemorrhage is not the major cause of death in GCRV infection”. Does it mean that NO production may not be the main agent involved in GCRV-induced pathogenesis?

Abbreviations should be listed in alphabetical order.

Round 2

Reviewer 2 Report

The manuscript has been significantly improved over the previous version.

Only a few minor issues:

In figue S3 caption, please indicate the concentration of SMT compound, and the time after GCRV challenge when the samples were collected.

line 137: SMT is a selective inhibitor.

line 157: SMT directly inhibits
